# Population Genetics of *Anopheles pretoriensis* in Grande Comore Island

**DOI:** 10.3390/insects14010014

**Published:** 2022-12-23

**Authors:** Melina Campos, Nikita Patel, Carly Marshall, Hans Gripkey, Robert E. Ditter, Marc W. Crepeau, Ali Toilibou, Yssouf Amina, Anthony J. Cornel, Yoosook Lee, Gregory C. Lanzaro

**Affiliations:** 1Vector Genetics Laboratory, Department of Pathology, Microbiology, and Immunology, University of California, Davis, CA 95616, USA; 2Malaria Control Program, Moroni, Comoros; 3Mosquito Control Research Laboratory, Kearney Research and Extension Center, Department of Entomology and Nematology, University of California, Parlier, CA 93648, USA; 4Florida Medical Entomology Laboratory, Department of Entomology and Nematology, Institute of Food and Agricultural Sciences, University of Florida, 200 9th St SE, Vero Beach, FL 32962, USA

**Keywords:** island biogeography, population genetics, secondary vector, malaria

## Abstract

**Simple Summary:**

In this study, we assessed phylogenetics of 24 *Anopheles* species and the genetic structure of *Anopheles pretoriensis* populations on Grande Comore Island. Our study is the first to report the whole mitochondria genome of *A. pretoriensis* and the first to inform on the genetic relationship of this species’ populations, both within the island of Grande Comore and between the island and continental Africa. Studies on secondary vectors of malaria, such as *A. pretoriensis*, are significant because they have been reconsidered in regard to their role in sustaining malaria transmission after primary vectors are controlled.

**Abstract:**

*Anopheles pretoriensis* is widely distributed across Africa, including on oceanic islands such as Grande Comore in the Comoros. This species is known to be mostly zoophylic and therefore considered to have low impact on the transmission of human malaria. However, *A. pretoriensis* has been found infected with *Plasmodium*, suggesting that it may be epidemiologically important. In the present study, we sequenced and assembled the complete mitogenome of *A. pretoriensis* and inferred its phylogenetic relationship among other species in the subgenus *Cellia*. We also investigated the genetic structure of *A. pretoriensis* populations on Grande Comore Island, and between this island population and sites in continental Africa, using partial sequence of the mitochondrial cytochrome *c* oxidase subunit I (*COI*) gene. Seven haplotypes were found on the island, one of which was ubiquitous. There was no clear divergence between island haplotypes and those found on the continent. The present work contributes knowledge on this understudied, yet abundant, *Anopheles* species.

## 1. Introduction

Malaria is a life-threatening disease caused by protozoa in the genus *Plasmodium* and is transmitted to humans by female mosquitoes of the genus *Anopheles*. There are 465 *Anopheles* species officially recognized worldwide, only 70 of which are known to transmit human malaria parasites [1]. These species differ in geographic distribution and can be classified as primary vectors, those that contribute the most to malaria transmission in a specific area, or as alternative vectors, those with secondary importance in malaria epidemiology. The latter are often overlooked, but studies have shown that these species may sustain malaria transmission after primary vector populations are suppressed or eliminated [2,3,4]. 

Malaria is endemic in the Union of the Comoros, a nation composed of three volcanic islands in the Indian Ocean: Grande Comore, Mohéli, and Anjouan. In 2010, the country reached its record in malaria cases, when over 100,000 cases were reported [5]. In the following years, a combination of preventative and control measures caused a substantial decline in malaria incidence [6]. These efforts placed Grande Comore Island within the WHO control phase in the path toward malaria-free status and placed the remaining islands in the pre-elimination phase [5,7].

Seven species of *Anopheles* have been reported to be present in the Comoros: the primary vectors include *A. gambiae s.s.* (hereafter *A. gambiae*), *A. merus*, and *A. funestus*; and secondary vector/non-vector species present are *A. coustani*, *A. maculipalpis*, *A. mascarensis* and *A. pretoriensis* [5,8,9]. It is unknown if the latter species play a significant role in malaria transmission in the Comoros. However, *A. coustani* and *A. pretoriensis* have been found to host *P. falciparum* sporozoites elsewhere in Africa [2,4,10]. Here, we present a new mitogenome assembly for *A. pretoriensis* and describe the genetic structure of populations on Grande Comore Island and their relationship with neighboring mainland populations. This study improves the state of knowledge of this potential malaria vector.

## 2. Materials and Methods

### 2.1. Site Description

Grande Comore is the largest (1148 sq. km) and youngest island in the Comoros archipelago which lies roughly midway between Madagascar and Mozambique in the Indian Ocean. The island has a tropical climate with a hot and rainy season (rainfall ranging from 1500 to 2000 mm) occurring between November and April and a cool and dry season (rainfall ranging from 500 to 800 mm) between May and October. There is an active volcano, Mount Karthala (2361 m), in the south-central part of the island that forms a belt of higher elevation (>500 m) running north to south. Watercourses are absent in Grande Comore due to the highly permeable volcanic substrate which led to widespread construction of large water cisterns for domestic use. These cisterns are the main breeding sites for mosquitoes on the island (reviewed in [5]). Grande Comore is home to the capital city of Moroni and is the most populated of the three islands that constitute the Union of the Comoros.

### 2.2. Mosquito Sampling

Immature stages of mosquitoes were collected from 20 sites from Grande Comore in March 2020 (Figure 1, Appendix A). Breeding sites were predominantly in outdoor cisterns followed by pools of standing water in roads and near villages. A subset of samples was reared to the adult stage, and these were morphologically identified to species in the field. All samples were preserved individually in 1.5 ml tubes filled with 80% ethanol and archived at the Vector Genetics Laboratory, at the University of California, Davis. Individual mosquito DNA was extracted using a Qiagen Biosprint machine following our established protocol [11]. Molecular identification was performed using the ITS2 region following an established protocol [2].

### 2.3. Mitogenome Sequencing and Analysis

The whole sequence of *A. pretoriensis* mitochondria was assembled using total genomic sequencing from one female adult specimen. Following extraction DNA yield was measured using a dsDNA high sensitivity assay kit on a Qubit instrument (Thermo Fisher Scientific, Waltham, MA, USA). The KAPA HyperPlus Kit (Roche Sequencing Solutions, Indianapolis, IN, USA) was used for library preparation using 10 ng DNA as input, as described in [12]. Library size selection and clean-up was performed using AMPure SPRI beads (Beckman Coulter Life Sciences, Indianapolis, IN, USA). The library was sequenced for 150 bp paired-end reads using a HiSeq 4000 instrument (Illumina, San Diego, CA, USA) at Novogene Corporation, Sacramento, CA. Raw-sequencing reads were used to assemble the mitogenome using NOVOPlasty version 2.6.7 [13]. The final contig was annotated with MITOS [14] using the invertebrate genetic code under default settings. 

Phylogenetic analyses included the newly assembled *A.* (*Cellia*) *pretoriensis* mitogenome, 20 additional species also in the subgenus Cellia, and three species, each from a different subgenus: *Anopheles* (*A. coustani*), *Nyssorhyncus* (*A. darlingi*) and *Kerteszia* (*A. cruzii*) (accession numbers in Appendix A). Protein coding sequences were used for multiple alignment of mitogenomes using Muscle v3.8 [15] built-in Geneious Prime 2022.2 (https://www.geneious.com, accessed on 28 November 2022). Phylogenetic trees were constructed using maximum likelihood (ML) inference with 1000 bootstrap replicates using a general time reversible (GTR) mode in PhyML 3.3 [16]. 

### 2.4. COI Sequencing and Analysis

Partial sequence of the mitochondrial cytochrome *c* oxidase subunit I (*COI*) gene was amplified and sequenced from 84 *A. pretoriensis* specimens from Grande Comore Island (Appendix A). Universal primer pair (LCO1490: 5′-GGTCAACAAATCATAAAGATATTGG-3′; HCO2198: 5′-TAAACTTCAGGGTGACCAAAAAATCA-3′) was used to amplify this region [17]. Amplification of *COI* was performed in a 25 μl PCR mixture consisting of 10 pmol of each primer and 12.5 μl of GoTaq^®^ Green Master Mix (Promega Corporation, Madison, USA) and 1 μl of DNA template. Thermocycler conditions were as follows: 94 °C for 5 min; followed by 30 cycles of 94 °C for 40 s, 50 °C for 1 min, and 72 °C for 1 min; and a final extension step of 72 °C for 5 min. PCR products were visualized by 1.5% agarose gel electrophoresis. Positive amplicons were then purified using AMPure SPRI beads (Beckman Coulter Life Sciences, Indianapolis, IN, USA), following manufacturers protocol. Amplicons were sent for Sanger sequencing at the UC Davis DNA Sequencing facility.

*COI* sequences were combined in a multiple sequence alignment using Muscle v3.8 [15] with default parameters and used to build a haplotype network using TCS network in PopArt [18]. Genetic diversity within *A. pretoriensis* populations in Grande Comore Island was determined by haplotype diversity (*Hd*) and nucleotide diversity (π) as estimated using DnaSP v6 [19]. An additional 10 *A. pretoriensis COI* sequences from mainland Africa (Ethiopia, Kenya, Mozambique, Malawi, South Africa, and Zambia) were retrieved from GenBank and included in a ML phylogenetic tree using GTR mode in PhyML 3.3 [16], with 1000 bootstrap replications.

## 3. Results

### 3.1. Mosquito Identification and Abundance

A total of 923 specimens were collected, preserved, and molecularly identified from 20 sites in Grande Comore Island. Two species of *Anopheles* were identified: *A. pretoriensis* and *A. gambiae*. *Anopheles pretoriensis* was ubiquitous on the island and accounted for 81.6% of specimens collected during this study (Figure 1, Appendix A). *Anopheles gambiae* was present in 8 of the 20 sites sampled. Its distribution was limited to southern and central western sites in Grande Comore (Figure 1). There was a difference of about 70 bp in the length of ITS2 region between *A. gambiae* and *A. pretoriensis*, confirmed by Sanger sequence (accession numbers OP851349 and OP851350, respectively); therefore, we used ITS2 PCR for species identification.

### 3.2. Anopheles Pretoriensis Mitogenome Phylogeny

The complete mitogenome of *A. pretoriensis* (Genbank: ON479654) has a length of 15,712 bp. Thirteen protein coding genes (PCGs), 22 tRNA genes and 2 rRNA genes were annotated in the mitogenome and its AT content was 77.2%, which is comparable to other species in the subgenus *Cellia* [20]. 

The phylogenetic analysis (Figure 2) contains 24 terminals from the genus *Anopheles*, representing four subgenera (*Kerteszia*, *Nyssorhyncus*, *Anopheles* and *Cellia*) with four series within the subgenus *Cellia* (Myzomyia, Neocellia, Neomyzomyia and Pyretophorus). Relationships among subgenera and series within the subgenus *Cellia* were resolved with significant support using ML analysis (ML > 70) and these were for the most part concordant with conventional morphological classifications. However, relationships within the series Neocellia and Neomyzomyia were unresolved. *Anopheles* (*Kerteszia*) *cruzii* was used to root the phylogeny. *Anopheles* (*Nyssorhyncus*) *darlingi* was recovered as the earliest branching lineage and is sister to a clade of subgenera *Anopheles* plus *Cellia* (Figure 2). Subgenus *Anopheles* was recovered as a sister clade to *Cellia*. The series Pyretophorus was recovered as the earliest branching lineage within the *Cellia* clade and is a monophyletic sister clade to Myzomyia + Neocellia + Neomyzomyia (Figure 2). The series Neomyzomyia was recovered as a sister clade to Neocellia + Myzomyia (Figure 2). Myzomyia was recovered as a monophyletic clade and is sister to the clade of Neocellia (Figure 2).

### 3.3. Anopheles pretoriensis Population Genetics

Following DNA extraction and ITS2 PCR molecular diagnostic, specimens of *A. pretoriensis* the *COI* fragment were sequenced (GenBank accession on Appendix A). All *COI* sequences matched (>99% percent identity) previously reported *A. pretoriensis* sequences deposited in NCBI (as of 27 May 2022). Analysis of *COI* sequences from 84 *A. pretoriensis* specimens yielded 7 haplotypes, distinguished by 9 informative variable sites and 1 singleton (Figure 3A). One haplotype, H1, was present in all but one site (Bangoi Hambou) in Grande Comore Island (Figure 3B). The overall haplotype diversity (*Hd*) and nucleotide diversity (π) of *A. pretoriensis* were 0.56 and 0.0025, respectively.

The ML phylogeny constructed using the partial *COI* gene region (final length of 611 bp) represents 20 terminals from the genus *Anopheles* and three species: *A. gambiae*, *A. stephensi* and *A. pretoriensis*. *Anopheles pretoriensis* specimens included 7 unique haplotypes from Grande Comore Island and sequences retrieved from GenBank from continental Africa: Ethiopia, Kenya, Malawi, Mozambique, South Africa and Zambia. Relationships among *A. pretoriensis* samples within the *COI* phylogeny were mostly unresolved using ML analysis (ML > 70, Figure 3C). *Anopheles gambiae* from Grande Comore Island (OP620446) was used to root the phylogeny and we recovered *A. stephensi* (MN329060) from India as sister to the *A. pretoriensis* clade. We recovered two individual *A. pretoriensis* (MK628505 and MK625502) as the earliest branching lineages to the unresolved polyphyletic clade comprised of the remaining *A. pretoriensis* (Figure 3C).

## 4. Discussion

*Anopheles pretoriensis* was first described in 1903 by Theobald [21]. The species name is based on the location of its discovery in Pretoria, South Africa. This species’ geographic distribution is widespread in Africa, being present in over 30 continental countries [22,23] and in islands such as Cape Verde [24], Comoros [8], Madagascar [25] and Mayotte [26]. *Anopheles pretoriensis* has also been reported in Saudi Arabia [27] and Yemen [28]. Taxonomic classification places *A. pretoriensis* within the series Neocellia in the subgenus *Cellia*. Our phylogenetic analysis is concordant with this taxonomic arrangement, grouping *A. pretoriensis* with other members of Neocellia series included here (*A. maculatus*, *A. splendidus* and *A. stephensi*). 

In the Comoros, *A. pretoriensis* has been found on all three islands as reported by a field expedition in 1949 by Lavergne [29]. Similarly, our collection in Grande Comore identified the presence of only two anophelines: *A. gambiae* and *A. pretoriensis*. Marsden et al. [8] reported consistent differences in wing markings between *A. pretoriensis* specimens found in the Comoros and continental Africa, indicating local divergence not detected in our *COI* sequence analysis. Seven *COI* haplotypes were uniquely found in Grande Comore; however, our phylogenetic analysis revealed unsupported divergence between those haplotypes and the ones from mainland Africa. There was no population structure of *A. pretoriensis* on Grande Comore Island, with haplotype H1 being predominant (present in 17 out of 18 sites). Genetic diversity was lower than other species from mainland [10,30,31] as expected under island biogeography theory [32]. The most ancestral population were from the continental African country of Ethiopia, suggesting an east African origin of *A. pretoriensis*.

*Anopheles pretoriensis* has been generally considered as a non-vector of human malaria, due to its zoophylic behavior [22]. However, a recent study detected positive *Plasmodium* infections in several *Anopheles* species considered as non-vectors, including *A. pretoriensis* [2]. This suggests that control strategies that suppress or eliminate the major vector, *A. gambiae*, may result in the elevation of *A. pretoriensis* to major malaria vector status on Grande Comore. Additional studies on *Plasmodium* infectivity would expand our knowledge of the vectorial competence and capacity of *A. pretoriensis*. The present study was the first to sequence and assemble the mitogenome of *A. pretoriensis*, as well as to conduct population genetics investigation of this species on Grande Comore Island. 

## Figures and Tables

**Figure 1 insects-14-00014-f001:**
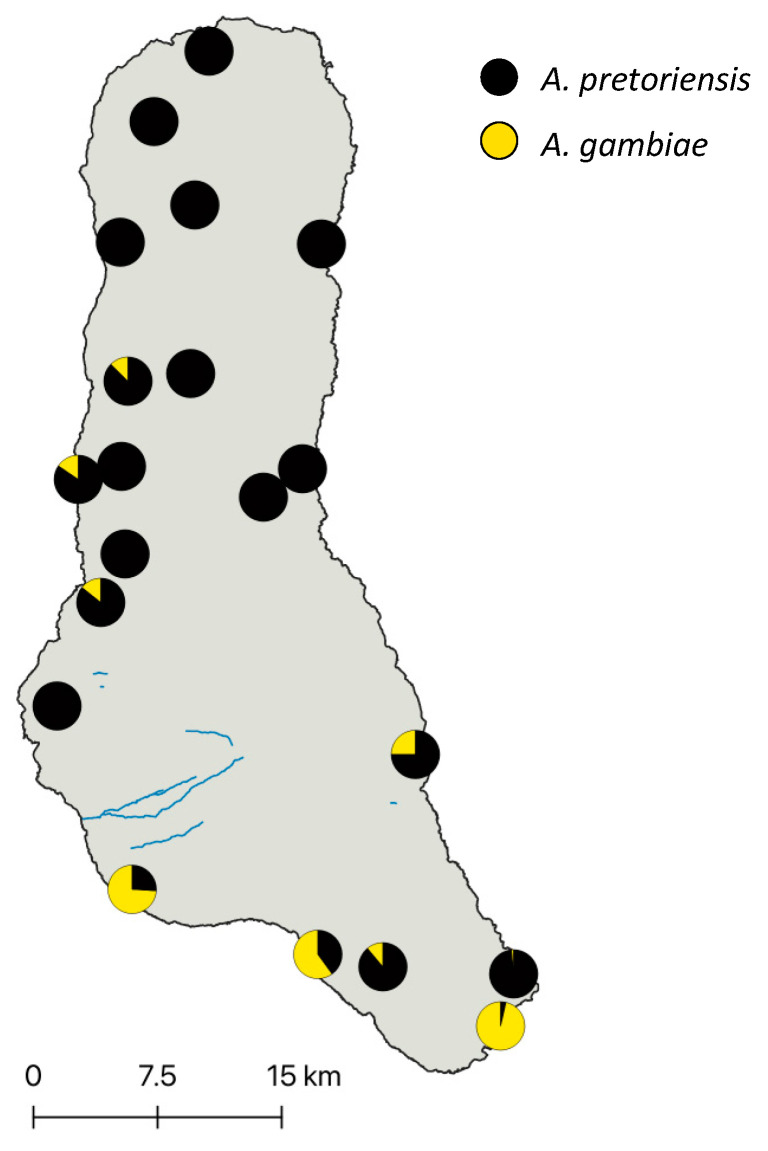
Relative abundance of *Anopheles* spp. in Grande Comore Island. Map of sampling sites where immature specimens of *Anopheles* were collected across Grande Comore. Pie charts display relative abundance of *A. gambiae* (yellow) and *A. pretoriensis* (black).

**Figure 2 insects-14-00014-f002:**
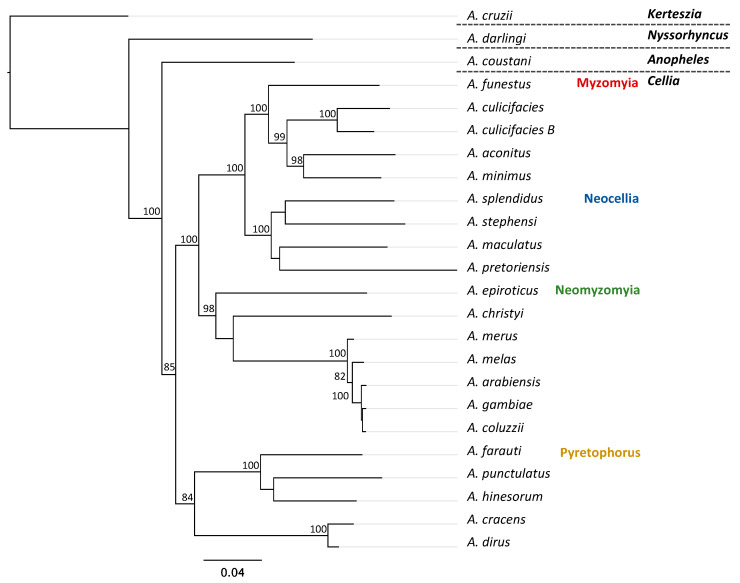
Phylogenetic relationships of *Anopheles* species based on mitogenomes. Maximum-likelihood phylogeny was constructed with sequence of 13 protein coding genes from 21 species of *Anopheles* from the four series within the subgenus *Cellia*, including the newly sequenced *A. pretoriensis*. One species from each of the subgenera: *Anopheles*, *Nyssorhyncus* and *Kerteszia*. Branch values indicate bootstrap replications (displayed values > 80).

**Figure 3 insects-14-00014-f003:**
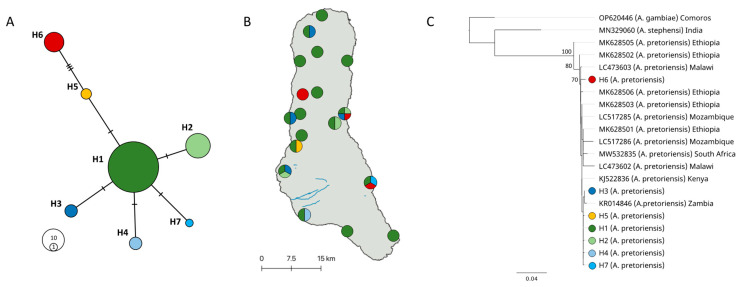
Population genetics of *A. pretoriensis* in Grande Comore Island. (**A**) Haplotype network of *COI* sequences from 84 *A. pretoriensis* specimens collected throughout Grande Comore. Haplotypes were named H1-H7 and color coded. (**B**) Geographic distribution of *COI* haplotypes; haplotype H1 was present in all but one site. (**C**) Maximum-likelihood phylogenetic tree of the 7 *COI* haplotypes found in Grande Comore and 12 *COI A. pretoriensis* sequences from specimens collected from continental Africa. *Anopheles gambiae* and *A. stephensi* were included as outgroup species. Branch values indicate supported bootstrap values (>70).

## Data Availability

The data presented in this study are openly available in GenBank: mitochondrial genome (accession number ON479654); COI sequences (accession on Appendix A).

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
