# Peer review of "Population Genetics of Anopheles pretoriensis in Grande Comore Island"

_insects, 2022, doi:10.3390/insects14010014_

Round 1
Reviewer 1 Report
I congratulations the authors for the article. Here I make some comments and observations.
On line 46 - The species A.gambiae is a Complex of cryptic species. The species A.merus is part of this complex. Is the species A.gambiae recorded here sensu stricto? Do other species of the Gambiae Complex occurs int he study area? I believe it is important to inform if only these two species of the Complex occur in the study area.
On line 94 - It is not necessary to write A.(subgenu.Cellia)pretoriensis, just write A.(Cellia)pretoriensis and the peers understand that the species belongs to the subgenus Cellia.
On line 145, 146 and147 I would write ...The species A.cruzii belonging to the subgenus Kerteszia was used to root the phylogeny. The species A.darlingi of the subgenu Nyssorhynchus was recovered as the earlist branching lineage and is sister to a clade of sugenera Anopheles plus Cellia. Or simply...The species A.(Kerteszia) cruzii was used to root the phylogeny. The species A.(Nyssorhynchus) darlingi was recovered as the......because it is already written to which subgenus the two species belong.
On line 146 - correct ..."A.(subgenu. Nyssorhynchus) darlingi ....to A.(Nyssorhynchus)darlingi
Lines 212 and 213 of the discussion "This suggests that control strategies that suppress or eliminate the major vector, A.gambiae may result in the elevation of A.pretoriensis to major malaria vector status". Certainly other aspects need to be studied to verify whether A.pretoriensis could become the primary vector of malaria in the context that the authors discuss. In adittion to isolating the malaria parasite from females of the species, studies are important thar show the vectorial competence of the species. That is whether the species is able to become infected and live long enough to pass the product of this infection to the hosts. Infectivity and longevity studies are very important for A. pretoriensis.
Figure 1 shows the relative abundande of A.pretoriensis and its wide territorial distribution,. If the species proliferates in areas densely populated by the human population these would also be aspects that favor the species to be a googd vector.
There are the comments I have to make
Congratulations.
Author Response
I congratulations the authors for the article. Here I make some comments and observations.
On line 46 - The species A.gambiae is a Complex of cryptic species. The species A.merus is part of this complex. Is the species A.gambiae recorded here sensu stricto? Do other species of the Gambiae Complex occurs int he study area? I believe it is important to inform if only these two species of the Complex occur in the study area.
The text was corrected to A. gambiae s.s. which is the species we are referring in the manuscript. Sister species have been found in other islands in the Comoros that are not being considered here.
On line 94 - It is not necessary to write A.(subgenu.Cellia)pretoriensis, just write A.(Cellia)pretoriensis and the peers understand that the species belongs to the subgenus Cellia.
The text was corrected accordingly.
On line 145, 146 and147 I would write ...The species A.cruzii belonging to the subgenus Kerteszia was used to root the phylogeny. The species A.darlingi of the subgenu Nyssorhynchus was recovered as the earlist branching lineage and is sister to a clade of sugenera Anopheles plus Cellia. Or simply...The species A.(Kerteszia) cruzii was used to root the phylogeny. The species A.(Nyssorhynchus) darlingi was recovered as the......because it is already written to which subgenus the two species belong.
The text was corrected accordingly.
On line 146 - correct ..."A.(subgenu. Nyssorhynchus) darlingi ....to A.(Nyssorhynchus)darlingi
The text was corrected accordingly.
Lines 212 and 213 of the discussion "This suggests that control strategies that suppress or eliminate the major vector, A.gambiae may result in the elevation of A.pretoriensis to major malaria vector status". Certainly other aspects need to be studied to verify whether A.pretoriensis could become the primary vector of malaria in the context that the authors discuss. In adittion to isolating the malaria parasite from females of the species, studies are important thar show the vectorial competence of the species. That is whether the species is able to become infected and live long enough to pass the product of this infection to the hosts. Infectivity and longevity studies are very important for A. pretoriensis.
We added the following statement in the text: “Additional studies on Plasmodium infectivity would illuminate the vectorial competence and capacity of A. pretoriensis.”
Figure 1 shows the relative abundande of A.pretoriensis and its wide territorial distribution,. If the species proliferates in areas densely populated by the human population these would also be aspects that favor the species to be a googd vector.
There are the comments I have to make
Congratulations.
Reviewer 2 Report
I hope, like a classic entomologist, not to expect to control mosquitoes with genetic work. Of course, in parallel with genetic studies, it is necessary to ensure that the mosquitoes do not move much from their breeding site. we don't know much about the changes we cause by applying the results of our studies. A. pretoriensis is a zoophilic species and, as a great Thai researcher who identified hundreds of species in the 1960s said, that if there weren't the bites of zoophilic mosquitoes, many animals in the forests would die. Good job without presumption but with prudence
Author Response
I hope, like a classic entomologist, not to expect to control mosquitoes with genetic work. Of course, in parallel with genetic studies, it is necessary to ensure that the mosquitoes do not move much from their breeding site. we don't know much about the changes we cause by applying the results of our studies. A. pretoriensis is a zoophilic species and, as a great Thai researcher who identified hundreds of species in the 1960s said, that if there weren't the bites of zoophilic mosquitoes, many animals in the forests would die. Good job without presumption but with prudence.
We appreciate the time you took to read our work.
Reviewer 3 Report
Greetings and Regards
The main research question
Population genetics of Anopheles pretoriensis in Grande Comore Island
The introduction is properly described, I checked, but it needs more explanation.
And due to the importance of investigating and identifying the Anopheles pretoriensis species, this article is suitable for publication in the insect journal.
According to the authors, there was no clear divergence between island haplotypes and those found on the continent. The present work provides knowledge about this understudied yet abundant Anopheles species. And in general, it is very important in the world of entomology.
Figure 3 is not clear and needs further explanation.
In terms of written language, it needs some changes.
In general, the article is suitable for publication with a few changes
It is suggested to compare this species in other countries in the future.
Author Response
Greetings and Regards
The main research question
Population genetics of Anopheles pretoriensis in Grande Comore Island
The introduction is properly described, I checked, but it needs more explanation.
And due to the importance of investigating and identifying the Anopheles pretoriensis species, this article is suitable for publication in the insect journal.
According to the authors, there was no clear divergence between island haplotypes and those found on the continent. The present work provides knowledge about this understudied yet abundant Anopheles species. And in general, it is very important in the world of entomology.
Figure 3 is not clear and needs further explanation.
The caption was edited to further explain the figure.
In terms of written language, it needs some changes.
English was carefully reviewed and revised as necessary.
In general, the article is suitable for publication with a few changes
It is suggested to compare this species in other countries in the future.